# The cryo-EM structure of ASK1 reveals an asymmetric architecture allosterically modulated by TRX1

**Karolina Honzejkova[1], Dalibor Kosek[2], Veronika Obsilova[2]\*, Tomas Obsil[1,2]\***

[1]Department of Physical and Macromolecular Chemistry, Faculty of Science, Charles University, Prague, Czech Republic; [2]Institute of Physiology of the Czech Academy of Sciences, Laboratory of Structural Biology of Signaling Proteins, Division BIOCEV, Vestec, Czech Republic

**\*For correspondence:**
veronika.obsilova@fgu.cas.cz
(VO);
obsil@natur.cuni.cz (TO)

**Competing interest:** The authors declare that no competing interests exist.

**Abstract** Apoptosis signal-regulating kinase 1 (ASK1) is a crucial stress sensor, directing cells toward apoptosis, differentiation, and senescence via the p38 and JNK signaling pathways. ASK1 dysregulation has been associated with cancer and inflammatory, cardiovascular, and neurodegenerative diseases, among others. However, our limited knowledge of the underlying structural mechanism of ASK1 regulation hampers our ability to target this member of the MAP3K protein family towards developing therapeutic interventions for these disorders. Nevertheless, as a multidomain Ser/Thr protein kinase, ASK1 is regulated by a complex mechanism involving dimerization and interactions with several other proteins, including thioredoxin 1 (TRX1). Thus, the present study aims at structurally characterizing ASK1 and its complex with TRX1 using several biophysical techniques. As shown by cryo-EM analysis, in a state close to its active form, ASK1 is a compact and asymmetric dimer, which enables extensive interdomain and interchain interactions. These interactions stabilize the active conformation of the ASK1 kinase domain. In turn, TRX1 functions as a negative allosteric effector of ASK1, modifying the structure of the TRX1-binding domain and changing its interaction with the tetratricopeptide repeats domain. Consequently, TRX1 reduces access to the activation segment of the kinase domain. Overall, our findings not only clarify the role of ASK1 dimerization and inter-domain contacts but also provide key mechanistic insights into its regulation, thereby highlighting the potential of ASK1 protein-protein interactions as targets for anti-inflammatory therapy.

## eLife assessment

This **important** manuscript reports the cryo-EM structure of the ASK1 protein, which is a critical regulator of the MAPKs, JNKs, and p38 MAPKs in diverse cellular stress responses. The evidence of ASK1 interaction with TRX1 is **compelling** and will eventually allow the discovery of small molecule inhibitors of ASK1 activity.

## Introduction

Mitogen-activated protein (MAP) kinase cascades are one of the most important signal transduction networks in cells. Highly conserved throughout eukaryotes (*Widmann et al., 1999*), MAP kinase pathways are activated in response to a number of stimuli, such as cytokines, growth factors, oxidative stress, calcium influx, and lipopolysaccharides, promoting proliferation, inflammatory responses, and apoptosis (*Cuevas et al., 2007*). The incoming signals are transmitted across a common, three-layered protein kinase system composed of the upstream MAP kinase kinase kinase (MAP3K), the intermediate MAP kinase kinase (MAP2K), and the downstream MAP kinase (MAPK) (*Widmann et al.,*

*1999*). MAP2Ks and MAPKs are similarly activated through phosphorylation by an upstream kinase, but MAP3K activation is much more complex. MAP3Ks require a tight regulation given the high number of their potential triggers and the serious consequences of their dysregulation, in the form of various diseases (*Fujino et al., 2007*; *Peti and Page, 2013*).

Cancer and inflammatory, cardiovascular, and neurodegenerative diseases, among others, have been extensively associated with excessive ASK1 signaling, in particular (*Katome et al., 2013*; *Ma et al., 2019*; *Meijles et al., 2020*; *Okazaki, 2017*). Also known as MAP3K5, ASK1 stands out for its role in directing cells toward apoptosis, differentiation, and senescence via the p38 and JNK MAP kinase pathways (*Ichijo et al., 1997*; *Sawada et al., 2001*). However, currently available JNK and p38 inhibitors lack efficacy and/or have undesirable side effects (*Bühler and Laufer, 2014*; *Ijaz et al., 2009*). Therefore, as their upstream activator, ASK1 is a prospective target for therapeutic intervention in these disorders, especially considering its wide range of triggers and interactions.

Various stress stimuli, such as oxidative and endoplasmic reticulum (ER) stress and calcium influx, activate ASK1, thus explaining why this Ser/Thr-specific protein kinase is such a crucial stress sensor (*Ichijo et al., 1997*). In line with this function, human ASK1 is a multi-domain protein consisting of an N-terminal thioredoxin-binding domain (TBD), a central regulatory region (CRR) formed by tetratrico-peptide repeats (TPR), and pleckstrin-homology (PH) domains, a kinase domain (KD), and a C-ter-minal coiled-coil (CC) region followed by a sterile alpha motif (SAM) domain (*Figure 1A*; *Bunkoczi et al., 2007*; *Psenakova et al., 2020*; *Trevelyan et al., 2020*; *Weijman et al., 2017*). In both resting and active states, ASK1 forms a large multiprotein complex, the ASK signalosome, with dozens of interacting proteins, including other ASK family members ASK2 and ASK3 (*Federspiel et al., 2016*; *Kaji et al., 2010*; *Saitoh et al., 1998*; *Subramanian et al., 2004*). In the resting state, ASK1 constitu-tively oligomerizes, presumably via the C-terminal CC region and the SAM domain, remaining inac-tive through interactions with TRX1, glutaredoxin and 14-3-3 proteins (*Bunkoczi et al., 2007*; *Fujino et al., 2007*; *Kosek et al., 2014*; *Liu et al., 2000*; *Petrvalska et al., 2016*; *Psenakova et al., 2020*; *Saitoh et al., 1998*; *Subramanian et al., 2004*; *Tobiume et al., 2002*; *Trevelyan et al., 2020*).

Although the exact role of these binding partners in ASK1 regulation is still debated, TRX1 binding to the N-terminal TBD is thought to prevent homophilic interactions between ASK1 N-termini containing the TBD, TPR, PH, and KD domains (*Fujino et al., 2007*; *Weijman et al., 2017*). When complexed with TRX1, inactive ASK1 is ubiquitinated and degraded (*Liu and Min, 2002*). By contrast, under oxidative stress, TRX1 oxidation triggers its dissociation from the ASK signalosome, followed by 14-3-3 disso-ciation and tumor necrosis factor receptor-associated factor (TRAF2/5/6) recruitment to CRR (*Fujino et al., 2007*; *Goldman et al., 2004*; *Gotoh and Cooper, 1998*; *Kylarova et al., 2016*; *Nadeau et al., 2009*; *Nadeau et al., 2007*; *Saitoh et al., 1998*; *Noguchi et al., 2005*; *Psenakova et al., 2020*; *Song and Lee, 2003*). Presumably through CRR or another ASK1 domain N-terminal to KD, death domain-associated protein 6 (Daxx) directly interacts with ASK1, thereby mediating ASK1 activation by the death-inducing ligand system, known as cluster of differentiation 95 (CD95)/Fas receptor (*Chang et al., 1998*). These events likely enable homophilic interactions between ASK1 N-termini, leading to Thr838 phosphorylation in the activation segment either by trans-autophosphorylation or by protein serine/threonine kinase 38 (MPK38) and subsequent ASK1 activation (*Jung et al., 2008*; *Liu et al., 2000*; *Nadeau et al., 2007*; *Tobiume et al., 2002*).

Combined, these findings have advanced our knowledge of the structure of individual ASK1 domains (*Bunkoczi et al., 2007*; *Psenakova et al., 2020*; *Trevelyan et al., 2020*; *Weijman et al., 2017*). Yet, several questions as to how these domains interact with each other, how they participate in ASK1 oligomerization, or what role these interactions play in ASK1 regulation remain unanswered. Moreover, the mechanism whereby TRX1 binding to the N-terminal TBD inhibits ASK1 activation is also unresolved. CRR may keep TBD and KD relatively close, enabling TRX1 to block CRR and/or KD and, hence, preventing substrate binding and inhibiting ASK1 (*Weijman et al., 2017*). But ASK1 dimerization most likely significantly affects not only the conformation of ASK1 but also its interdomain interactions. Considering the above, overcoming current obstacles to the development of effective drug therapies targeting ASK1 requires acquiring relevant structural data to further our understanding of the complex regulation of ASK1.

To gain structural insights into ASK1 regulation and the role of TRX1 binding, this study aims at structurally characterizing the dimeric, C-terminally truncated ASK1 with the domains TBD, TPR, PH, and KD in a state close to its active form and its complex with TRX1. For this purpose, we used several

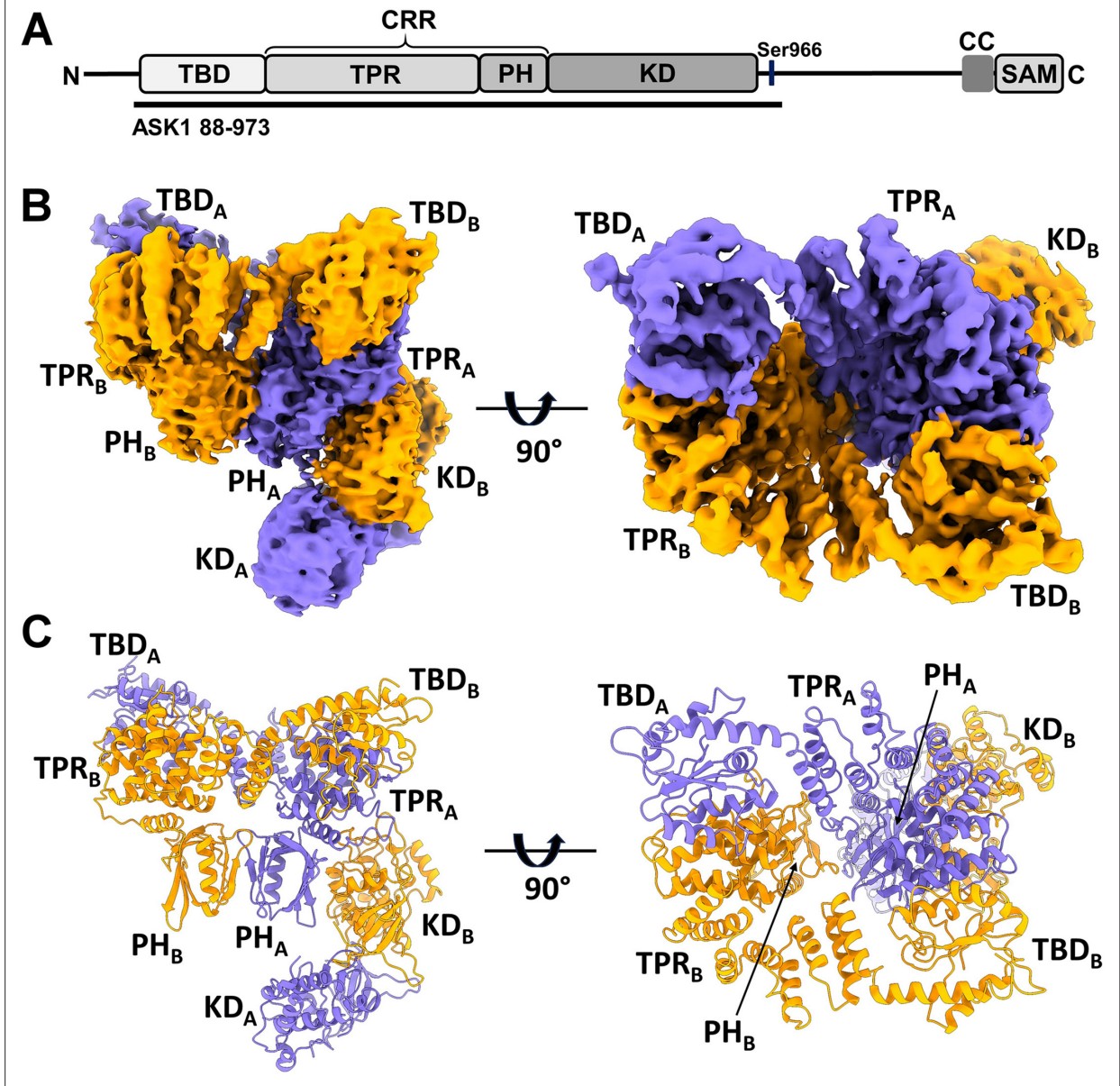

**Figure 1.** Structure of C-terminally truncated apoptosis signal-regulating kinase 1 (ASK1). (**A**) Schematic domain structure of ASK1. TBD, thioredoxin-binding domain; TPR, tetratricopeptide repeats; PH, pleckstrin-homology domain; CRR, central regulatory region; KD, kinase domain; CC, coiled-coil motif; SAM, sterile alpha motif domain. Black bar represents the construct used in cryo-electron microscopy (cryo-EM) analysis. (**B, C**) Cryo-EM density map and cartoon view of C-terminally truncated dimeric ASK1. Density maps were generated using threshold level 5.5.

The online version of this article includes the following figure supplement(s) for figure 1:

**Figure supplement 1.** Apoptosis signal-regulating kinase 1 (ASK1) TBD-CRR-KD data processing.

**Figure supplement 2.** Apoptosis signal-regulating kinase 1 (ASK1) TBD-CRR-KD map quality assessment.

**Figure supplement 3.** Agreement between the experimental density map and the refined model of C-terminally truncated apoptosis signal-regulating kinase 1 (ASK1).

biophysical methods, including cryo-electron microscopy (cryo-EM), hydrogen/deuterium exchange coupled to mass spectrometry (HDX-MS) and sedimentation velocity analytical ultracentrifugation (SV AUC). Our data reveal that ASK1 forms a compact and asymmetric dimer in which all four N-terminal domains are involved in extensive interdomain and interchain interactions. These interactions stabilize the active conformation of the ASK1 kinase domain. TRX1 functions as a negative allosteric effector of ASK1 by modulating the structure of all its N-terminal domains, including the activation segment

of the catalytic domain. Overall, our findings not only clarify the role of ASK1 dimerization and inter-domain contacts but also provide key mechanistic insights into its regulation.

## Results

### ASK1 forms an asymmetric and compact dimer through extensive inter-domain and inter-chain interactions

Given the low expression yield and solubility of full-length human ASK1, we designed a C-terminally truncated construct consisting of TBD, CRR, and KD (residues 88–973), thus all domains crucial for ASK1 regulation. The expression yield and stability of this protein was sufficient for subsequent studies. Cryo-EM imaging of the ASK1 TBD-CRR-KD revealed well-dispersed particles, with 2D class averages showing obvious secondary structure elements (*Figure 1—figure supplement 1*). Approximately 5780 micrograph movies enabled single-particle reconstructions of this protein at a nominal resolution of 3.7 Å, as further detailed in Methods and *Supplementary file 1* and *Figure 1—figure supplement 2*.

The cryo-EM map revealed that C-terminally truncated ASK1 forms a compact and asymmetric dimer, enabling extensive interdomain and interchain interactions (*Figure 1B and C* and *Figure 1—figure supplement 3*). One side of the molecule is formed by TBD and TPR domains of both protomers, with TBD domains embedded between TPR domains. The TPR domains then interact with a dimer of PH domains, and the resulting dimeric TBD-TPR-PH module has a funnel shape with approximately twofold rotational symmetry. The other side of the molecule is formed by a KD dimer, which interacts with the N-terminal TBD-TPR-PH module through the KD of only one protomer. The binding site of this KD is located at the interface between the TPR and PH domains of the opposite protomer, and the second KD has no contact with the N-terminal domains of either its own or the other chain.

### All N-terminal domains participate in ASK1 homodimerization

The region of the cryo-EM map that corresponds to TBD domains (residues 95–266) was interpreted using the AlphaFold model of ASK1 (AF-Q99683-F1). This model suggested that TBD consists of a six-stranded, mostly parallel, β-sheet decorated with several α-helices, thus resembling the thioredoxin structure. Such a conformation was in line with our cryo-EM map, which revealed a similar arrangement of secondary structure elements in both TBDs (*Figure 2A* and *Figure 2—figure supplement 1*). TBD is a compact and globular domain that interacts with the TPR domain of the opposite protomer via the helix α3 and the loop between α4 and β5. The C-terminal helix α5 of TBD is connected by a short loop to helix α6, which is the first helix of the TPR domain. In the same chain, there are no other contacts between TBD and TPR except for the interaction between the C-terminus of helix α3 of TBD and the N-terminus of helix α6 of TPR.

In a previous study, cysteine C250, located at the N-terminus of α5, was identified as a crucial residue for TBD structural integrity and TRX1 binding (*Kosek et al., 2014*; *Psenakova et al., 2020*). Corroborating these findings, our structure of TBD$_A$ suggests that this cysteine residue forms a disulfide bridge with cysteine C225 located nearby (*Figure 2A*). This disulfide, whose formation was shown previously (*Kylarova et al., 2016*), likely stabilizes the region containing residues 216–245 located between β5 and α5. The quality of the cryo-EM map in this region is worse for TBD of the opposite chain (TBD$_B$), but here too C225 and C250 are close to each other. In addition, this region appears to be very flexible as no interpretable density was found for residues 228–245 in both TBDs. This result is consistent with a previous NMR characterization of isolated TBD, which showed that this domain retains substantial conformational plasticity (*Psenakova et al., 2020*).

The conformation of both CRR domains resembles the crystal structure of isolated CRR, with 14 tightly arranged α-helices (α6-α19) forming seven tetratricopeptide repeats (TPRs), followed by a PH domain (*Weijman et al., 2017*; *Figure 1C* and *Figure 2—figure supplement 1*). Superpositions with the crystal structure showed minor helix shifts in the first four repeats of both TPRs, most likely due to interactions with the TBD of the opposite chain (*Figure 2—figure supplement 2*). Both CRRs interact directly only through the β-sheets of their PH domains, and the resulting CRR dimer has twofold rotational symmetry (*Figure 1B and C*). Combined, extensive interactions between N-terminal TBD, TPR, and PH domains emerge as a crucial factor for the homodimerization of C-terminally truncated ASK1.

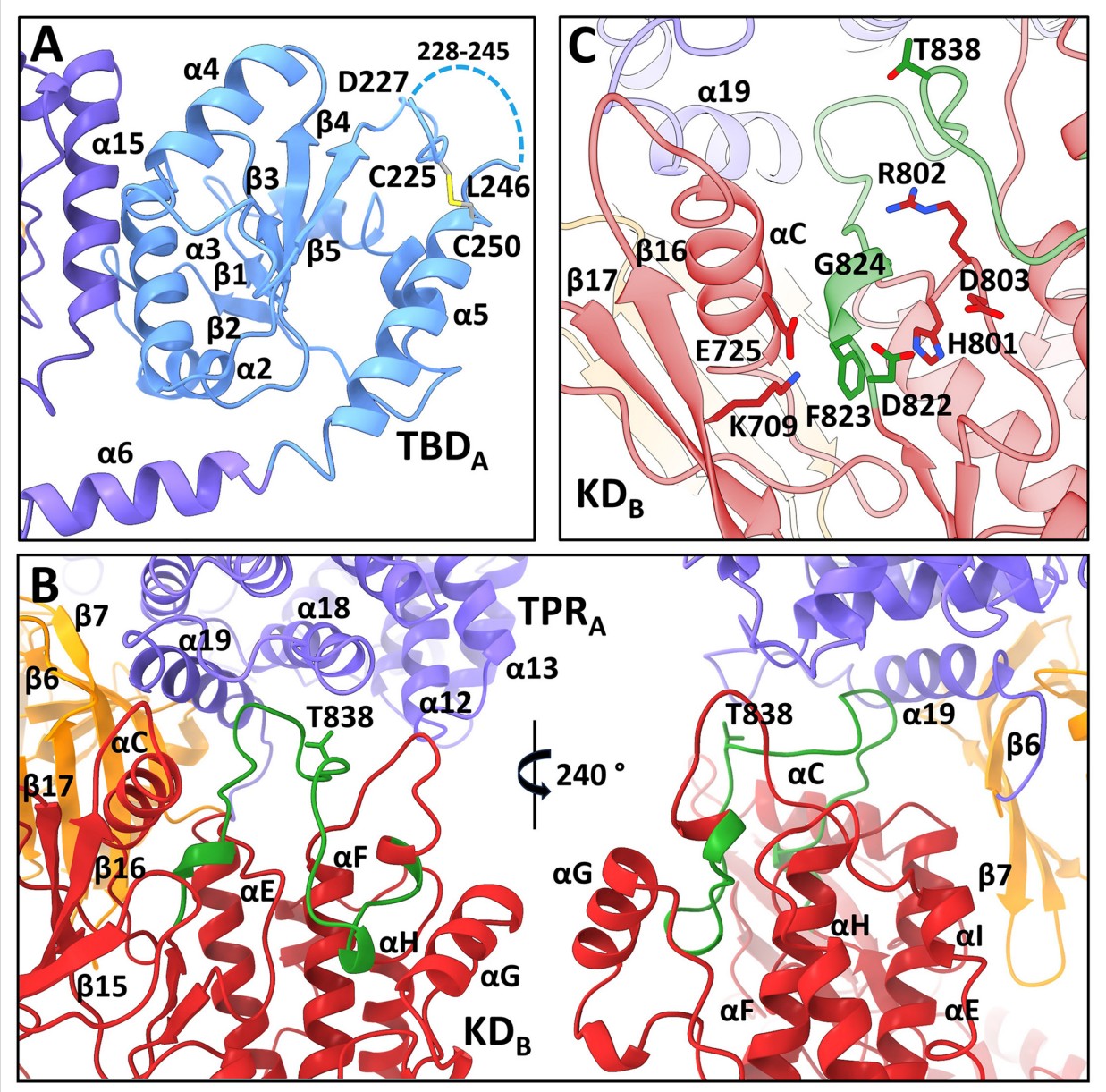

**Figure 2.** All four N-terminal domains of apoptosis signal-regulating kinase 1 (ASK1) are involved in extensive interdomain and interchain interactions.
(**A**) Cartoon representation of TBD$_A$ and its interaction with helix α15 of TPR$_B$. The dashed line indicates the missing section (residues 228–245).
(**B**) Cartoon representation of interactions between KD$_B$, TPR$_A$, and PH$_A$ domains (shown in red, violet, and orange, respectively). The activation segment is shown in green. (**C**) Detailed view of the active site of KD$_B$. KD$_B$ is shown in red, TPR$_A$ is shown in violet. The activation segment is shown in green.

The online version of this article includes the following figure supplement(s) for figure 2:

**Figure supplement 1.** Secondary structure of apoptosis signal-regulating kinase 1 (ASK1) TBD-CRR-KD.

**Figure supplement 2.** Comparison of central regulatory region (CRR) regions with the crystal structure of the isolated CRR.

**Figure supplement 3.** Comparison of kinase domains (KDs) with the crystal structure of the isolated KD.

## Interdomain interactions stabilize the activation segment of the kinase domain

TPR and PH domains of one chain (chain A) create a docking platform for the KD of the opposite chain (KD$_B$), which interacts with TPRs 4 (helices α12, α13) and 7 (helices α18, α19) of TPR$_A$, and β6 and β7 strands of PH$_A$, mainly via its C-lobe (*Figure 2B*). The kinase domains of both protomers dimerize, as previously observed in the crystal structure of isolated ASK1 KD with bound inhibitor staurosporine

(*Bunkoczi et al., 2007*), i.e., in a head-to-tail orientation through an extensive interface spanning almost the entire length of the domain. However, its comparison with the crystal structure showed changes in the position of the αC (α21) helix, which is in the inward active position, in both KDs (*Figure 2—figure supplement 3A and B*). This shift positions the conserved active site lysine (K709) within the hydrogen-bond distance of the αC glutamate (E725), as usually found in active kinases (*Johnson et al., 1996*; *Figure 2C* and *Figure 2—figure supplement 3E*).

The cryo-EM map of $KD_B$, which interacts with $TPR_A$ and $PH_A$ domains, enabled us to build the whole activation segment (residues 822–849). This segment adopts a conformation competent for substrate binding (*Figure 2B*). A similar conformation was observed in the crystal structure of ASK1 KD, which, however, showed only the beginning and end of the activation segment (*Figure 2—figure supplement 3B*; *Bunkoczi et al., 2007*). In this conformation, the conserved residue E725 from the αC (α21) helix is within the hydrogen-bond distance of the main chain of the DFG motif residues F823 and G824 at the beginning of the activation segment (*Figure 2C* and *Figure 2—figure supplement 3E*).

A similar conformation of the activation segment was also observed in the active form of another MAP3K BRAF (*Figure 2—figure supplement 3C*; *Haling et al., 2014*). In active BRAF, the conformation of the activation segment is stabilized by E611 interactions with R575 from the catalytic HRD motif, but in the $KD_B$ of the present structure, the activation segment is stabilized by interactions with the α18 and α19 helices from the last TPR repeat of the $TPR_A$ domain (*Figure 2B*).

The activation segment of the KD domain from the opposite chain ($KD_A$), which has no contact with N-terminal domains, is not visible in our map, likely due to its high flexibility. However, the beginning and end of this segment adopt a conformation similar to that observed in the crystal structure of the isolated ASK1 KD (*Bunkoczi et al., 2007*). Together with the inward position of the helix αC (α21), this conformation suggests that $KD_A$ is also in an active conformation (*Figure 2—figure supplement 3A*).

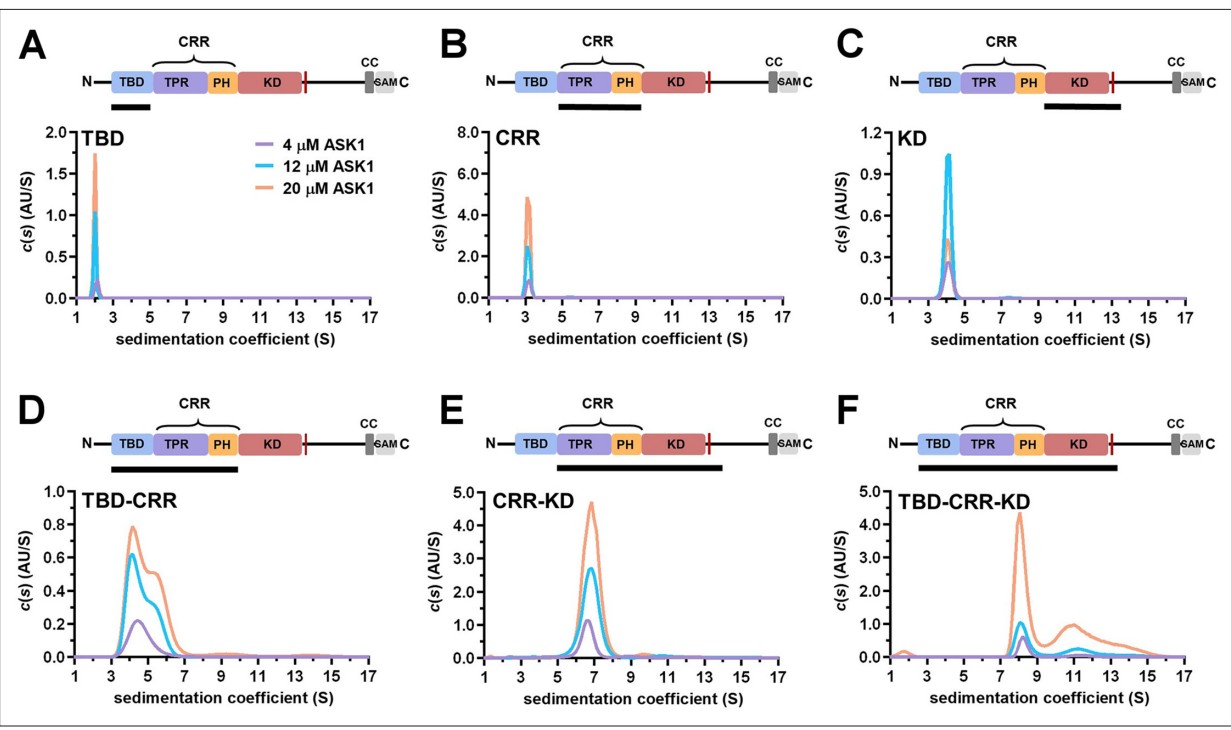

**Figure 3.** TBD-CRR forms dimers in solution, but not TBD or CRR. (**A–F**) Sedimentation coefficient distributions (*c*(*s*)) of different apoptosis signal-regulating kinase 1 (ASK1) N-terminal constructs (TBD, CRR, KD, TBD-CRR, CRR-KD, and TBD-CRR-KD) measured at concentrations of 4 μM, 12 μM and 20 μM. Schematic domain structure of ASK1 is shown at the top; the black bar represents the used construct. TBD, thioredoxin-binding domain; TPR, tetratricopeptide repeats; PH, pleckstrin-homology domain; CRR, central regulatory region; KD, kinase domain; SAM, sterile alpha motif domain.

The online version of this article includes the following figure supplement(s) for figure 3:

**Figure supplement 1.** Sedimentation velocity analytical ultracentrifugation (SV AUC) analysis shows the effect of thioredoxin 1 (TRX1) binding on the dimerization of apoptosis signal-regulating kinase 1 (ASK1) TBD-CRR-KD.

The comparison with the crystal structure of isolated KD revealed a change in the relative position of the KD subunits. The superimposition of the KD dimer of ASK1 TBD-CRR-KD with the crystal structure of the KD dimer (*Bunkoczi et al., 2007*) through $KD_A$ is shown in *Figure 2—figure supplement 3D*. $KD_B$ is slightly rotated relative to the equatorial plane of the dimer due to interactions of its C-lobe with $TPR_A$. This rotation results in a~4 Å shift of the C-lobe relative to its position in the crystal structure of the KD dimer. Overall, interdomain interactions within ASK1 TBD-CRR-KD stabilize the conformation of the activation segment. As a result, the activation segment remains competent for substrate binding.

## TBD-CRR, but neither TBD nor CRR form dimers in solution

Previous studies have shown that isolated KD forms a stable dimer with a $K_D$ of ~220 nM in which the protomers are oriented in a head-to-tail manner and interact through a large dimerization interface (*Bunkoczi et al., 2007*; *Petrvalska et al., 2016*). To verify the interdomain contacts observed in the cryo-EM map and their relevance to the homodimerization of C-terminally truncated ASK1, we prepared isolated TBD, CRR, and KD and two additional constructs composed of TBD-CRR and CRR-KD to study their oligomerization by sedimentation velocity analytical ultracentrifugation (SV AUC).

The sedimentation coefficient distribution $c(s)$ of the isolated ASK1 KD assessed by SV AUC showed only one peak with weight-average sedimentation coefficients corrected to 20.0 °C and to the density of water, $s_{w(20,w)}$, of 4.2 S (estimated $M_w$~62 kDa), most likely corresponding to the KD dimer (the theoretical $M_w$ of the KD dimer is 72.4 kDa) (*Figure 3C*). In contrast, the isolated domains, TBD and CRR, were protomeric in solution as their sedimentation coefficient distributions $c(s)$ showed peaks with $s_{w(20,w)}$ of 2.2 and 3.3 S with estimated $M_w$~16 kDa and ~38 kDa, respectively (theoretical $M_w$ of TBD and CRR are 20.6 and 45.5 kDa, respectively) (*Figure 3A and B*). While the ASK1 TBD-CRR construct showed concentration-dependent dimerization, as indicated by its bimodal $c(s)$ distribution (*Figure 3D*), the ASK1 CRR-KD construct formed a stable dimer in solution, with a $s_{w(20,w)}$ of 6.9 S (estimated $M_w$~166 kDa, theoretical $M_w$ 163 kDa) (*Figure 3D and E*). Similarly, as shown by its $c(s)$ distribution, the longest ASK1 TBD-CRR-KD construct also formed stable dimers in solution, with a $s_{w(20,w)}$ of 8.5 S (*Figure 3F*). This value is consistent with the theoretical sedimentation coefficient value of 9 S calculated using the HydroPRO program (*Ortega et al., 2011*) and the cryo-EM structure of the ASK1 TBD-CRR-KD dimer. The $c(s)$ distribution of ASK1 TBD-CRR-KD also revealed higher oligomers, whose abundance increased with the concentration.

The dimerization of KD-containing ASK1 constructs is not surprising because KDs form stable dimers, and CRRs do not interfere with the dimerization interface; on the contrary, they facilitate dimerization through interactions between PH domains (*Figure 1B and C*). Furthermore, the absence of dimerization of an isolated TBD is consistent with its interactions within the N-terminal TBD-CRR module, wherein TBDs interact only with TPR, not with each other. Isolated CRR showed no dimerization either, suggesting that contacts between PH domains observed in the ASK1 TBD-CRR-KD dimer are not strong enough to allow the formation of a stable CRR dimer and likely result from KD dimerization. Combined, our SV AUC measurements with isolated domains and their pairs are consistent with the cryo-EM structure of C-terminally truncated ASK1 and confirm that all three N-terminal domains are involved in homophilic interactions between ASK1 chains.

## TRX1 binding induces structural changes in the activation segment of ASK1 KD

TRX1 binding presumably inhibits ASK1 activation by blocking the homophilic interaction of the N-terminal part of ASK1, according to the currently accepted model of ASK1 activation under oxidative stress conditions (*Fujino et al., 2007*; *Saitoh et al., 1998*). Since we were unable to prepare a sufficiently stable ASK1:TRX1 complex for cryo-EM analysis, apparently due to their relatively weak interactions under the conditions used in this study (*Kosek et al., 2014*), we assessed TRX1 binding effects on ASK1 TBD-CRR-KD structure and dimerization by SV AUC and HDX-MS. Unexpectedly, SV AUC showed that TRX1 binding has no effect on ASK1 TBD-CRR-KD dimerization. However, when we compared $c(s)$ distributions, we found a significant reduction in peak area, in the region of sedimentation coefficients 10–12 S, in the presence of TRX1. This result indicates that TRX1 prevents the formation of higher oligomers (*Figure 3—figure supplement 1*).

By HDX-MS, we monitored the kinetics of hydrogen-to-deuterium exchange along the polypeptide backbone because this method enables us to evaluate the structure of proteins and their complexes (*Wales and Engen, 2006*). Hydrogens of amide groups involved in stable hydrogen bonds and/or sterically shielded from the solvent are protected from exchange. In contrast, flexible regions exposed to the solvent exchange more quickly than rigid and buried regions.

The comparison of ASK1 TBD-CRR-KD deuteration profiles with and without TRX1 revealed that several TBD regions were significantly (more than twice the standard deviation and above 3%) less deuterated, that is, more protected, upon TRX1 binding (*Figure 4A and B* and *Figure 4—figure supplements 1–3*). More specifically, significant protection was observed in residues 128–143, 184–195, and especially in segments 206–259. Many of these segments overlapped with regions previously identified by NMR as the TRX1 binding surface of TBD (*Psenakova et al., 2020*) and, moreover, included the flexible region between β5 and α5 containing C225 and C250 (*Figure 4B* and *Figure 4—figure supplements 2 and 3*). These regions on the flexible outer surface of the TBD near the interface between the TBD and the TPR of the opposite chain may form a TRX1-binding site.

This hypothesis is in line with the decrease in deuteration observed in the α15 helix of the TPR domain, which interfaces with TBD, in the adjacent α13, α14, and α19 helices, and in the α13-α14, α14-α15, α17-α18, and α18-α19 loops (*Figure 4A and B* and *Figure 4—figure supplement 4*). Within the PH domain, the β-strand β9 and the β8-β9 loop, located near the α19 helix of the TPR, were also protected. Through interactions, these regions connect the TRX1-binding site of TBD with the TPR and PH regions that form the docking surface of one of the two KDs (*Figure 4—figure supplement 2*). Given the lower deuteration of these regions, TRX1 binding likely stabilizes their structure and/or reduces their solvent accessibility (e.g. through conformational change), as shown by changes in the deuteration of several KD regions.

TRX1 binding to TBD reduced deuteration near the active site and at the C-terminus of the KD (*Figure 4A and B* and *Figure 4—figure supplement 5*). The most protected area included residues 830–844, which form the activation segment and interact with the TPR domain. In addition to the activation segment, the region containing αEF (α24), the helix αG (α26) and the αD-αE (α22-α23), αF-αG (α25-α26), and αG-αH (α26-α26) loops also showed reduced deuteration. Accordingly, these regions likely interact with each other, forming a coherent area whose structure and/or access to solvent is affected by TRX1 binding.

Significant protection was also observed in residues 655–670, which form a linker connecting PH and KD (*Figure 4A*). This linker should be quite flexible as no interpretable density was found in this region, in either protomer. However, TRX1 binding reduced its deuteration, suggesting structural changes at the interface of PH and KD. Taken together, our HDX-MS results indicate that TRX1 binding to TBD significantly alters the interactions and structure of TBDs, leading to changes in interactions within CRRs and to conformational changes in KDs, including in their activation segment and the C-terminus.

## TRX1 interacts with ASK1 through its active site

Based on comparisons of TRX1 deuteration profiles with and without ASK1 TBD-CRR-KD, the helix α2 regions and the β-strands β1, β3, and β4 (*Figure 4C and D* and *Figure 4—figure supplements 6 and 7*) are significantly protected. This protection includes the highly conserved catalytic motif W31CGPC35, located at the N-terminus of the α2 helices, whose sulfhydryl groups are responsible for TRX-dependent redox activity (*Holmgren et al., 1975*). This result is in line with previous findings according to which TRX1 binds to ASK1 through its active site (*Fujino et al., 2007*; *Kylarova et al., 2016*; *Psenakova et al., 2020*; *Saitoh et al., 1998*).

## Discussion

Homophilic interactions between N-terminal domains are crucial for ASK1 activation, as shown by our structural analysis of the C-terminally truncated ASK1. In addition to these interactions, ASK1 oligomerization is mediated by its C-terminal CC motif because the C-terminally truncated ASK1 has lower Thr838 phosphorylation levels and basal activity (*Tobiume et al., 2002*). Furthermore, the ASK1 construct with the first 384 residues of this protein, containing TBD and the N-terminal portion of TPR, can also oligomerize, as previously demonstrated by co-immunoprecipitation (*Fujino et al., 2007*) and

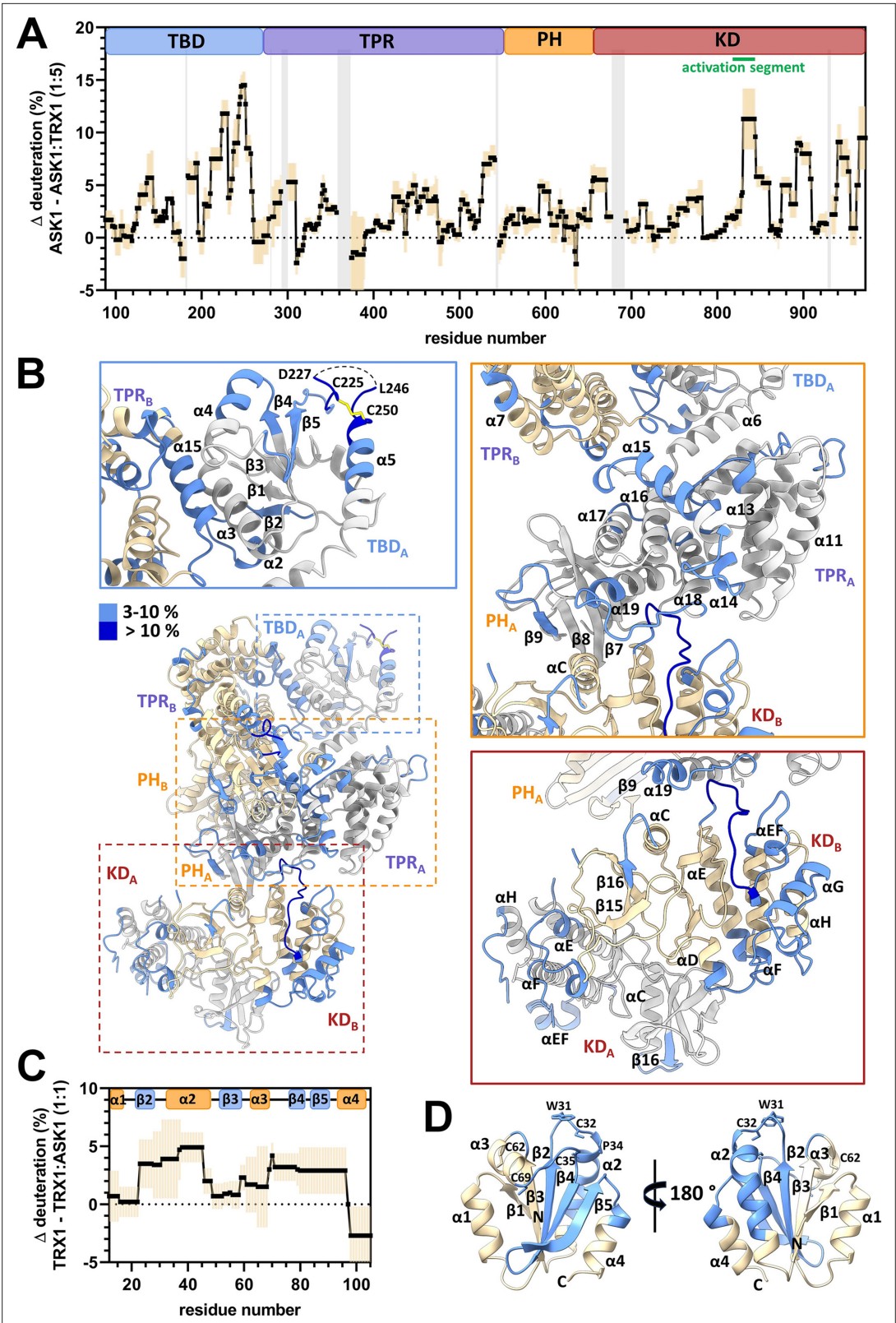

**Figure 4.** Thioredoxin 1 (TRX1) binding induces structural changes in all N-terminal domains of apoptosis signal-regulating kinase 1 (ASK1).
(**A**) Differences in ASK1 TBD-CRR-KD deuteration with and without TRX1 after 600 s. Positive values indicate protection (lower deuterium uptake) after TRX1 binding. The graph shows the average of three replicates (black points) and the standard deviation (light orange). Gray zones indicate areas without coverage. The domain structure of ASK1 TBD-CRR-KD is shown at the top. TBD, thioredoxin-binding domain; TPR, tetratricopeptide repeats;

*Figure 4 continued on next page*

*Figure 4 continued*

PH, pleckstrin-homology domain; CRR, central regulatory region; KD, kinase domain. (**B**) Cartoon representation of the structure of the ASK1 TBD-CRR-KD dimer (chain A in white, chain B in light yellow) colored according to changes in deuteration in the presence of TRX1 after 600 s. Changes in deuteration greater than twice the standard deviation and above 3% were considered significant. The insets show detailed views of the $TBD_A$ and its interface with $TPR_B$ (blue box), $CRR_A$ (orange box), and KD dimer (brown box). (**C**) Differences in TRX1 deuteration with and without ASK1 TBD-CRR-KD after 600 s. Positive values indicate protection (lower deuterium uptake) after ASK1 TBD-CRR-KD binding. The graph shows the average of three replicates (black points) and the standard deviation (light orange). The secondary structure of TRX1 is shown at the top. (**D**) Cartoon representation of TRX1 structure (in the reduced state, PDB ID: 1ERT [***Weichsel et al., 1996***]) colored according to changes in deuteration after ASK1 TBD-CRR-KD binding for 600 s.

The online version of this article includes the following figure supplement(s) for figure 4:

**Figure supplement 1.** Sequence coverage of apoptosis signal-regulating kinase 1 (ASK1) TBD-CRR-KD (residues 88–973) assessed by hydrogen/deuterium exchange (HDX).

**Figure supplement 2.** Thioredoxin 1 (TRX1) binding induces structural changes in all domains of apoptosis signal-regulating kinase 1 (ASK1) TBD-CRR-KD.

**Figure supplement 3.** Effect of thioredoxin 1 (TRX1) binding on thioredoxin-binding domain (TBD) deuteration in the apoptosis signal-regulating kinase 1 (ASK1) TBD-CRR-KD dimer.

**Figure supplement 4.** Effect of thioredoxin 1 (TRX1) binding on central regulatory region (CRR) deuteration in the apoptosis signal-regulating kinase 1 (ASK1) TBD-CRR-KD dimer.

**Figure supplement 5.** Effect of thioredoxin 1 (TRX1) binding on kinase domain (KD) deuteration in the apoptosis signal-regulating kinase 1 (ASK1) TBD-CRR-KD dimer.

**Figure supplement 6.** Sequence coverage of thioredoxin 1 (TRX1) assessed by hydrogen/deuterium exchange (HDX).

**Figure supplement 7.** Effect of complex formation on thioredoxin 1 (TRX1) deuteration.

in line with our SV AUC measurements (***Figure 3D***) and with our cryo-EM analysis of the apo form of ASK1 TBD-CRR-KD. This analysis revealed a compact and asymmetric dimer with all four N-terminal domains involved in extensive interdomain and interchain interactions. These interactions stabilize the active conformation of the kinase domain of ASK1 (***Figures 1 and 2***), highlighting their importance for ASK1 activation.

When comparing the conformation of KDs of the ASK1 TBD-CRR-KD dimer with the crystal structure of the isolated KD with bound inhibitor (***Bunkoczi et al., 2007***), we noted a shift in the αC helix, occupying an inward position typical of the active conformation (***Figure 2—figure supplement 3A and B***). Considering this position and the position of the activation segment, which is structured through interactions with the adjacent TPR and maintains a conformation competent for substrate binding, in the case of $KD_B$ (***Figure 2B and C***), interactions with the dimeric N-terminal TBD-CRR module may stabilize the active conformation of one of the two KDs. Furthermore, located in the activation segment and known to play a key role in ASK1 activation (***Tobiume et al., 2002***), T838 is oriented toward the solvent, so its (auto)phosphorylation could further stabilize the activation segment through interactions with residues (e.g. K526) of the loop between α18 and α19 of TPR located nearby.

In a recent study, we have shown that ASK1 TBD is structurally heterogeneous and has a globular conformation resembling the thioredoxin fold, with its C-terminal half (residues ~165–260) forming a TRX1-binding site (***Psenakova et al., 2020***). In this study, our cryo-EM structure revealed that these residues form not only the solvent-accessible outer surface of the ASK1 dimer but also the binding interface with the TPR domain (***Figure 2A***). Consequently, TRX1 binding significantly affects the structure of TBD and its interactions within the ASK1 dimer.

Our HDX-MS measurements after TRX1 binding revealed considerable changes in deuteration kinetics, in three regions of TBD, including its C-terminal half. TRX1 binding also affected the deuteration kinetics of several regions of the TPR and PH domains of CRR (***Figure 4A and B*** and ***Figure 4—figure supplements 2–4***), suggesting conformational changes, presumably a shift or change in the relative orientation of the TPR repeats, in this part of ASK1 (***Harrison and Engen, 2016***). Considering the differences between the TPR of the present ASK1 structure and the crystal structure of CRR (***Figure 2—figure supplement 2***; ***Weijman et al., 2017***) and its position in the ASK1 dimer, the ASK1 TPR domain appears to be a flexible structure. This structural flexibility enables communication between TBD and KD, as evidenced by structural changes in KD induced by TRX1 binding, and is in line with the role of TPR domains as structurally diverse scaffolding and binding modules, some

of which are quite flexible, triggering allosteric effects, and supporting molecular switch functions (*Perez-Riba and Itzhaki, 2019*; *Perez-Riba et al., 2018*). For example, allosteric effects regulate the binding of Hif, a TPR protein, to histone complexes H2A-H2B and H3-H4 (*Zhang et al., 2016*). Moreover, the flexibility or conformational plasticity of CRR may be involved in substrate recruitment (*Weijman et al., 2017*).

Many protein kinases are regulated through the phosphorylation of a residue(s) located in the activation segment (*Johnson et al., 1996*). For example, Akt1 kinase is autoinhibited through intra-molecular interactions between its PH and KD domains, limiting access to the activation segment (*Chu et al., 2020*). Our HDX-MS data suggest that ASK1 is regulated by a similar mechanism involving structural changes in the activation segment. As evidenced by the slower deuteration kinetics, TRX1 substantially reduces solvent access to this segment and/or decreases its flexibility by altering interactions either within KD and/or with TPR (*Figure 4A and B*), which may influence its phosphorylation on T838 within the signalosome. In doing so, TRX1 likely contributes to ASK1 inhibition (*Morita et al., 2001*; *Zhang et al., 2023*). Therefore, limiting access to residues of the activation segment regulated by phosphorylation may be thus a key regulatory event of ASK1 function.

In addition to the activation segment, TRX1 binding also decreased deuteration kinetics in other regions of KD, including the αG helix and surrounding loops (*Figure 4A and B*). These regions are adjacent to the activation segment, and in MAP3K BRAF, the helix αG is involved in binding to its substrate MEK1 (*Haling et al., 2014*). Accordingly, TRX1 may function as an allosteric effector, changing the conformation of key regions of the KD domain when binding to TBDs and, therefore, inducing conformational changes transferred via CRRs.

Rather than disrupting homophilic interactions of the N-terminal domain, TRX1 binding alters them, since no significant deprotection was detected by HDX-MS. These results were also consistent with SV AUC measurements. However, at intracellular ASK1 concentrations, which are substantially lower than those used in our SV AUC and HDX-MS experiments, TRX1 binding may reduce homophilic interactions of the N-terminal domains of ASK1, as previously shown by immunoprecipitation analysis (*Fujino et al., 2007*). The resulting shift in the equilibrium toward the protomer may explain this difference between our SV AUC and HDX-MS measurements and previously reported findings.

Inactive ASK1 is bound to scaffolding 14-3-3 proteins, which recognize a motif containing phosphorylated S966, located at the C-terminus of KD (*Goldman et al., 2004*; *Zhang et al., 1999*). 14-3-3 proteins suppress the catalytic activity of ASK1 through an unknown mechanism, albeit potentially involving suppression of homophilic interactions between N-terminal domains. In contrast, ASK1 activation requires TRAF2/5/6 or Daxx recruitment. These proteins interact with CRR and enhance N-terminal homophilic interactions of ASK1 protomers (*Chang et al., 1998*; *Fujino et al., 2007*; *Gotoh and Cooper, 1998*; *Saitoh et al., 1998*). Thus, TRAF2/5/6 or Daxx binding to CRR could stabilize the conformation of the ASK1 dimer, thereby promoting its activation.

Our structural analysis was performed with a C-terminally truncated ASK1 missing the CC motif and a C-terminal SAM domain, involved in ASK1 oligomerization. Thus, we cannot rule out the possibility that C-terminal segments may affect interactions of N-terminal domains, including KDs. In addition, the structures of relevant complexes must be solved in subsequent studies to elucidate in detail structural changes caused by TRX1 binding, as well as other binding partners, and to fully understand how these interactions contribute to ASK1 regulation. Notwithstanding these limitations, our data provide the first structural insights into C-terminally truncated ASK1 in a state close to its active form. ASK1 forms a compact and asymmetric dimer with all four N-terminal domains involved in extensive interdomain and interchain interactions that stabilize the active conformation of ASK1 KD. TRX1, a negative regulator of ASK1, functions as an allosteric effector. TRX1 binding affects the structure of TBD and its interaction with TPR, thereby affecting the structure of CRR and allosterically modulating KD, even reducing access to the activation segment with the key phosphorylation site T838 (*Figure 5*). Therefore, our findings open up opportunities for targeting (the) interaction(s) responsible for ASK1 activation towards developing selective ASK1 signaling inhibitors and ultimately pharmaceutical drugs for several inflammatory, cardiovascular, and neurodegenerative diseases, among others (*Budas et al., 2018*; *Ogier et al., 2020*). Moreover, these results should prompt further research on this key MAP3K and its regulation, with a significant translational output.

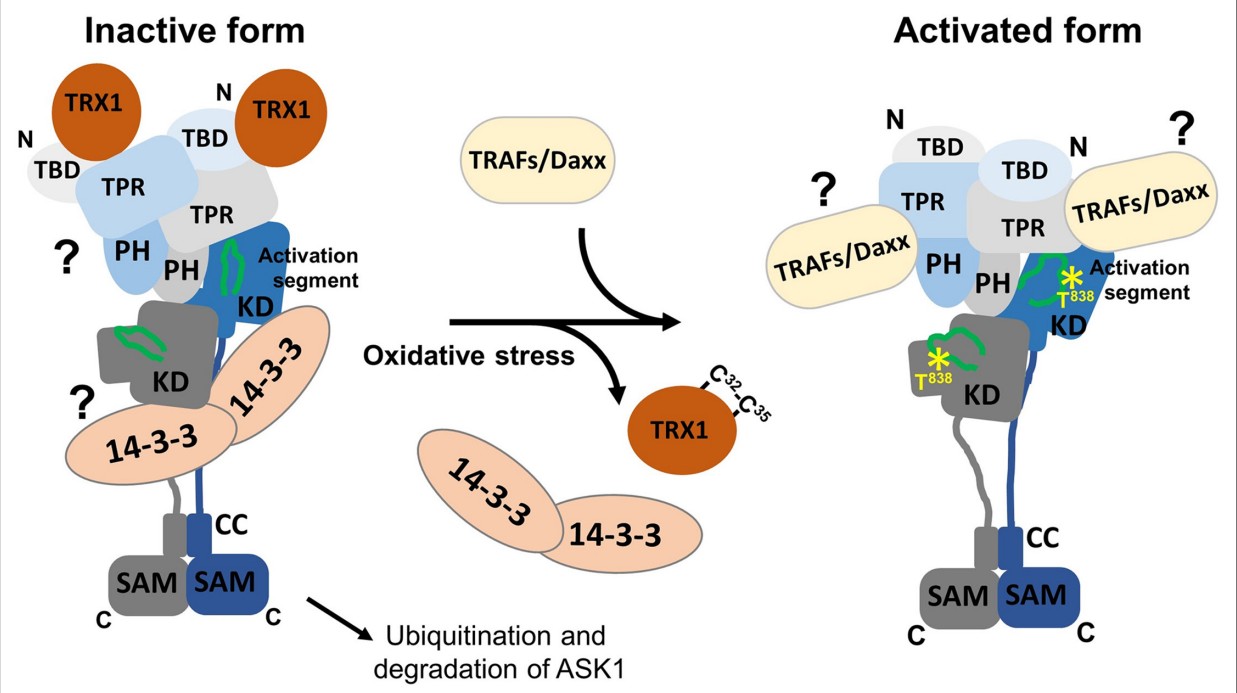

**Figure 5.** Proposed schematic model of apoptosis signal-regulating kinase 1 (ASK1) activation. In the resting state, ASK1 constitutively oligomerizes and is kept in an inactive state through interactions with thioredoxin 1 (TRX1) and 14-3-3 proteins. TRX1 appears to function as an allosteric effector whose binding affects the structure of thioredoxin-binding domain (TBD), likely affecting its interaction with tetratricopeptide repeats (TPR). Therefore, TRX1 affects the structure of central regulatory region (CRR) and allosterically modulates several regions of kinase domain (KD), even reducing access to the activation segment with the key phosphorylation site T838. Oxidative stress triggers TRX1 dissociation, followed by 14-3-3 dissociation and tumor necrosis factor receptor-associated factor (TRAF) recruitment. These events subsequently lead to a conformational change in KD and increase access to the activation segment, enabling its phosphorylation at T838 and, as a result, stabilizing the active conformation. The role of 14-3-3 proteins in ASK1 inhibition and the mechanism whereby TRAFs and Daxx are involved in ASK1 activation remain unclear and should be explored in subsequent studies.

## Materials and methods
### Recombinant protein expression and purification

Human TRX1 was expressed and purified as described previously (*Kosek et al., 2014*). To prevent TRX1 homodimerization caused by the formation of intermolecular disulfide bridges between non-active site C73 residues at high protein concentrations, we used the human TRX1 mutant C73S, which was reported to have unaltered structure and activity (*Forman-Kay et al., 1992*; *Weichsel et al., 1996*).

DNA encoding the TBD domain of human ASK1 (residues 88–267) was ligated into pRSFDuet-1 (Merck KGaA, Darmstadt, Germany) using BamHI and NotI sites. Modified pRSFDuet-1 containing the sequence of the 6×His-tagged GB1 domain of protein G inserted into the first multiple cloning site was kindly provided by Evzen Boura (Institute of Organic Chemistry and Biochemistry AS CR, Prague, Czech Republic). ASK1 TBD was expressed at 25 °C for 18 hr and purified from *Escherichia coli* BL21 (DE3) cells using Chelating Sepharose Fast Flow (GE Healthcare, Chicago, IL, USA) according to the standard protocol. The 6×His tag was cleaved by incubation with TEV protease (250 U of TEV/mg of fusion protein) at 4 °C overnight during dialysis against a buffer containing 50 mM Tris-HCl (pH 8), 0.5 M NaCl, 4 mM EDTA, 80 mM imidazole, 4 mM 2-mercaptoethanol, and 10% (w/v) glycerol. Chelating Sepharose Fast Flow (GE Healthcare, Chicago, IL, USA) was then used to capture the 6×His-GB1 and 6×His TEV, and the flow-through sample containing the ASK1 TBD protein was purified by size-exclusion chromatography on a HiLoad 26/600 Superdex 75 pg column (GE Healthcare, Chicago, IL, USA) in a buffer containing 20 mM HEPES (pH 7.0), 200 mM NaCl, 5 mM EDTA, 20 mM glycine, 5 mM DTT and 10% (w/v) glycerol.

TBD-CRR, CRR, KD, TBD-CRR-KD, and CRR-KD of human ASK1 (residues 88–658, 269–658, 658–973, 88–973, and 269–973, respectively) were ligated into the modified pRSFDuet-1 using BamHI

and NotI sites or pST39 using XbaI and BamHI sites. Proteins were expressed at 18 °C (ASK1 CRR-KD and ASK1 TBD-CRR-KD) or 25 °C (ASK1 TBD-CRR and ASK1 CRR) for 18 hr and purified from *Escherichia coli* BL21 (DE3) cells using Chelating Sepharose Fast Flow (GE Healthcare, Chicago, IL, USA) according to the standard protocol. Proteins were dialyzed overnight against a buffer containing 20 mM Tris-HCl (pH 7.5), 0.5 M NaCl, 1 mM EDTA, 2 mM 2-mercaptoethanol, and 10% (w/v) glycerol. ASK1 TBD-CRR and CRR constructs were incubated with TEV protease (250 U of TEV/mg of fusion protein) to remove 6×His-GB1 protein. ASK1 CRR-KD and TBD-CRR-KD contained uncleavable C-terminal 6×His tag. The final purification step was size-exclusion chromatography on a HiLoad 26/600 Superdex 75/200 pg column (GE Healthcare, Chicago, IL, USA) in a buffer containing 20 mM Tris-HCl (pH 7.5), 150 mM NaCl, 5 mM DTT, and 10% (w/v) glycerol. ASK1 KD (residues 658–973) was expressed and purified as described previously (*Petrvalska et al., 2016*).

## Analytical ultracentrifugation

Sedimentation velocity (SV) experiments were performed using a ProteomLabTM XL-I analytical ultracentrifuge (Beckman Coulter, Brea, CA, USA). Samples were dialyzed against a buffer containing 20 mM Tris-HCl (pH 7.5), 150 mM NaCl, and 2 mM 2-mercaptoethanol before the AUC measurements. SV experiments were conducted in charcoal-filled Epon centerpieces with a 12 mm optical path length at 20 °C and at rotor speeds ranging from 38,000–48,000 rpm (An-50 Ti rotor, Beckman Coulter). All sedimentation profiles were recorded with either absorption optics at 280 nm or interference optics. Buffer density and viscosity were estimated using the program SEDNTERP (*Laue et al., 1992*). Diffusion-deconvoluted sedimentation coefficient distributions $c(s)$ were calculated from raw data using the SEDFIT package (*Schuck, 2000*).

## Hydrogen/deuterium exchange coupled to mass spectrometry (HDX-MS)

ASK1 constructs (20 µM) and TRX1 were subjected to H/D exchange alone or in a mixture combining ASK1 TBD-CRR-KD with TRX1 (100 µM) and pre-incubated for 20 min at 4 °C. HDX reactions were performed by adding a 10×dilution of the protein mixture into a $D_2O$-based buffer containing 20 mM Tris-HCl (pD 7.5), 200 mM NaCl, 2 mM 2-mercaptoethanol, and 10% glycerol and by incubating at 4 °C. HDX was quenched after 2 s, 10 s, 1 min and 10 min of reaction by adding ice-chilled 1 M Glycine-HCl (pH 2.3), 6 M urea, 2 M thiourea, and 400 mM TCEP, in a 1:1 ratio, and all samples were frozen in liquid nitrogen. The 2 s and 10 min aliquots were prepared in triplicates. Thawed samples were loaded into the LC system including a custom-made pepsin/nepenthesin-2 protease column, and the generated peptides were online trapped and desalted on a SecurityGuard pre-column (ULTRA Cartridges UHPLC Fully Porous Polar C18, 2.1 mm, Phenomenex, Torrance, CA, USA) for 3 min under the flow of 0.4% formic acid (FA) in water, delivered at a flow rate of 200 µL.min$^{-1}$ (1260 Infinity II Quaternary pump, Agilent Technologies, Waldbronn, Germany). Desalted peptides were then separated on a reversed-phase analytical column (LUNA Omega Polar C18 Column, 100 Å, 1.6 µm, 100 mm × 1.0 mm, Phenomenex, Torrance, CA, USA) at a flow rate of 40 µL.min$^{-1}$ using a 10–40% linear gradient of solvent B (A: 2% acetonitrile/0.1% FA in water; B: 98% acetonitrile/0.1% FA in water) (1290 Infinity II LC system, Agilent Technologies, Waldbronn, Germany). The temperature within the customized LC system was kept at 0 °C to minimize the back exchange. All separated peptides were directly introduced into the ESI source of timsTOF Pro mass spectrometer with PASEF (Bruker Daltonics, Bremen, Germany). The data were analyzed using Data Analysis v. 5.3 (Bruker Daltonics, Bremen, Germany) and in-house DeutEx software. For each protein, peptides were identified by data-dependent LC–MS/MS using the same LC setup, performing a MASCOT (Matrix Science, London, UK) search against a custom-built database with sequences of ASK1, TRX1,and contaminants from the cRAP database.

## Cryo-EM - sample preparation and data collection

To prepare grids, thawed ASK1 TBD-CRR-KD was subjected to size exclusion chromatography on a Superdex 200 10/300 GL column (GE Healthcare, Chicago, IL, USA) in a buffer containing 20 mM Tris-HCl (pH 7.5), 150 mM NaCl, and 2 mM 2-mercaptoethanol. The peak fraction of ASK1 TBD-CRR-KD was diluted in a 1:1 ratio with a buffer containing CHAPSO to a final concentration of 3.9 mM CHAPSO and a final concentration of 0.9 mg/mL ASK1. Subsequently, 3.5 µL of protein solution was applied to a freshly glow-discharged (45 s total time, Gatan Solarus II 955 (Gatan, Inc, Pleasanton, CA,

USA)) UltrAuFoil holey grids (R1.2/1.3, Quantifoil, Großlöbichau, Germany). Blotting was performed using a Vitrobot Mark IV, for 4 s, at 20 °C and 100% humidity; all grids were plunge-frozen in liquid ethane and stored in liquid nitrogen until use. The grids were screened under a JEOL JEM 2100-plus electron microscope (Akishima, Tokyo, Japan) at 200 keV equipped with a TVIPS TemCam–XF416 4 K CMOS camera (TVIPS GmbH, Gauting, Germany) and under a Talos Arctica electron microscope (FEI, Thermo Fisher Scientific, Hillsboro, Oregon, USA) at 200 keV equipped with a GATAN K2 Summit detector (Gatan, Inc, Pleasanton, CA, USA). All data were collected under a Titan Krios electron microscope (FEI, Thermo Fisher Scientific, Hillsboro, Oregon, USA), at 300 keV, equipped with a Gatan K3 BioQuantum detector (Gatan, Inc, Pleasanton, CA, USA). Movies were recorded at 105,000x magnification and 0.834 Å per pixel calibrated resolution. The defocus values ranged from −0.7 to −2.8 μm, with a total exposure of 40 e⁻/Å². From 11,395 movies, 8691 were collected under a 40° tilt. Each movie consisted of 40 frames.

### Cryo-EM - image processing

All images were processed in CryoSPARC 4.1.2 (*Punjani et al., 2017*). Once the movies were imported and gain-corrected, they were subjected to patch motion correction and patch CTF estimation. Micrographs with an estimated CTF resolution >5 Å, full-frame motion distance >20 Å and a relative ice thickness >1.2 were discarded; then, micrographs were visually curated, and those with excessive aggregation, ice artifacts, or artifacts in power spectra were also excluded. The initial particle set was picked from 200 randomly selected micrographs, using a blob picker tool with a 60–180 Å particle size. After visual inspection, particles were extracted with a box size of 320 × 320 pixels, and after 2D classification, good classes were used to generate templates. Particle picking was then repeated using a template-based picker. Particle picks were extracted with a box size of 340 × 340 pixels and subjected to 2D classification. Particles within good 2D classes were used for Ab Initio 3D reconstruction, heterogeneous refinement, and homogeneous and non-uniform refinements with separated classes, including the calculation of gold standard Fourier shell correlation (GSFSC). GSFSC was used to determine the final map resolution with a 0.143 FSC threshold. The resulting map was sharpened using phenix.autosharpen (*Adams et al., 2019*). Details about data processing workflow are provided in *Supplementary file 1* and *Figure 1—figure supplement 1*.

### Cryo-EM - model building, refinement, and analysis

After visual inspection of the final map, we used the 'jiggle-fit' tool in Coot 0.9.8 (*Emsley and Cowtan, 2004*) to fit known crystal structures of different ASK1 domains, namely KD (PDB ID: 2CLQ [*Bunkoczi et al., 2007*]) and ASK1 CRR (PDB ID: 5ULM [*Weijman et al., 2017*]). Then, we fitted the Alpha-fold prediction (AF-Q99683-F1) to place a model for TBD. After finishing connections and adjustments, the model was excluded in areas of insufficient or uninterpretable density. Atomic refinement was performed using phenix.real_space_refine from the Phenix 1.19.1 software package (*Adams et al., 2019*). The model was validated using MolProbity (*Williams et al., 2018*). Model statistics are presented in *Supplementary file 1*. The final model has been deposited in PDB/EMDB under accession code: 8QGY/EMD-18396.

## Acknowledgements

This study was funded by the Czech Science Foundation (grant number 19-00121S), the Grant Agency of the Charles University (KH, grant number 1160120), and the Czech Academy of Sciences (RVO: 67985823 of the Institute of Physiology). We acknowledge CMS-Biocev ('Biophysical techniques, Crystallization, Diffraction, Structural mass spectrometry') of CIISB, and Cryo-electron microscopy and tomography core facility CEITEC MU of CIISB, Instruct-CZ Centres, supported by MEYS CR (LM2023042) and CZ.02.1.01/0.0/0.0/18_046/0015974. We thank P Pompach and P Vankova for their help with MS measurements, J Miksatko, J Novacek, and M Pinkas for their assistance with cryo-EM data collection, and Carlos V Melo for editing the article.

## Additional information

### Funding

| Funder | Grant reference number | Author |
|---|---|---|
| Grantová Agentura České Republiky | 19-00121S | Veronika Obsilova Tomas Obsil |
| Grantová Agentura, Univerzita Karlova | 1160120 | Karolina Honzejkova |
| Ministerstvo Školství, Mládeže a Tělovýchovy | LM2023042 | Tomas Obsil |
| European Regional Development Fund | CZ.02.1.01/0.0/0.0/18_046 /0015974 | Tomas Obsil |
| Czech Academy of Sciences | 67985823 | Veronika Obsilova Dalibor Kosek |

The funders had no role in study design, data collection and interpretation, or the decision to submit the work for publication.

### Author contributions

Karolina Honzejkova, Formal analysis, Investigation, Visualization, Writing - original draft, Writing - review and editing; Dalibor Kosek, Formal analysis, Investigation, Visualization, Methodology, Writing - original draft, Writing - review and editing; Veronika Obsilova, Tomas Obsil, Conceptualization, Supervision, Funding acquisition, Validation, Writing - original draft, Project administration, Writing - review and editing

### Author ORCIDs

Veronika Obsilova (iD) http://orcid.org/0000-0003-4887-0323
Tomas Obsil (iD) http://orcid.org/0000-0003-4602-1272

Reviewer #1 (Public Review): https://doi.org/10.7554/eLife.95199.2.sa1
Reviewer #2 (Public Review): https://doi.org/10.7554/eLife.95199.2.sa2
Author Response https://doi.org/10.7554/eLife.95199.2.sa3

## Additional files

### Supplementary files

• Supplementary file 1. Cryo-electron microscopy (Cryo-EM) data collection, refinement, and validation statistics.

• MDAR checklist

### Data availability

The authors declare that all data supporting the findings of this study are available within the article and its supplementary information file. Cryo-EM data have been deposited in the RCSB PDB/EMDB with the accession code: 8QGY/EMD-18396.

The following datasets were generated:

| Author(s) | Year | Dataset title | Dataset URL | Database and Identifier |
|---|---|---|---|---|
| Kosek D, Honzejkova K, Obsilova V, Obsil T | 2024 | Cryo-EM structure of C-terminally truncated Apoptosis signal-regulating kinase 1 (ASK1) | https://www.rcsb.org/ structure/8QGY | RCSB Protein Data Bank, 8QGY |
| Kosek D, Honzejkova K, Obsilova V, Obsil T | 2024 | Cryo-EM structure of C-terminally truncated Apoptosis signal-regulating kinase 1 (ASK1) | https://www.ebi.ac. uk/emdb/EMD-18396 | EMDataBank, EMD-18396 |

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
