## [Editor Report · eLife assessment]

This **important** manuscript reports the cryo-EM structure of the ASK1 protein, which is a critical regulator of the MAPKs, JNKs, and p38 MAPKs in diverse cellular stress responses. The evidence of ASK1 interaction with TRX1 is **compelling** and will eventually allow the discovery of small molecule inhibitors of ASK1 activity.

---

## [Referee Report · Reviewer #1 (Public Review)]

Summary:

Honzejkova K., et al. resolved the structure of one of the MAP3K proteins. Apoptosis signal-regulating kinase 1 (ASK1) is one of the main crucial stress sensors, which directs cells toward differentiation, and apoptosis. As a result, ASK1 dysregulation has been associated with a multitude of diseases like neurodegenerative, cardiovascular, and cancer. Understanding the structural-functional interplay of ASK1 would help researchers target this member of the MAP3K proteins to develop therapeutic interventions for these disorders.

Strengths:

Major strengths:

• Structure of the C-terminal truncated ASK1 protein.

Weaknesses:

• Lack of ASK1:TRX1 complex structure. The authors used instead SV AUC and HDX-MS techniques to compensate for the inability to get a sufficiently stable ASK1:TRX complex.

• There is not enough information about Cryo-EM data processing like 2D classification averages, local resolution of the EM map, or FSC figures.

• You can't reliably report the presence of a hydrogen bond with a 3.7Å resolution.

---

## [Referee Report · Reviewer #2 (Public Review)]

Summary:

The authors attempted to solve the 3D structure of ASK1 by Cryo-EM.

Strengths:

The authors solved the 3D structure of N-terminal domain s of ASK1 complexed with TRX. They found TRX1 functions as a negative allosteric effector of ASK1, modifying the structure of the TRX1-binding domain and changing its interaction with the tetratricopeptide repeats domain. The conclusions drawn from this paper are convincing and will greatly contribute to the development of new drugs targeting ASK1.

Weaknesses:

To study the ASK1 structure, C-terminally truncated ASK1 was used in the study, but not the full-length form of ASK1.

---

## [Author Response]

Our answer to reviewer #1 comments:

We attempted to perform structural characterization of the ASK1 complex with TRX1, but were unable to prepare a sufficiently stable ASK1:TRX1 complex for cryo-EM analysis, probably due to their relatively weak interactions. Therefore, we subsequently decided to use HDX-MS to characterize the structural changes of ASK1 induced by interactions with TRX1.

Detailed information about cryo-EM data processing including 2D classification averages, local resolution of the EM map and FSC figure are shown in Supporting Information, Supplementary Table S1 and Figures S1-S3.

We fully agree with the reviewer that the presence of hydrogen bonding cannot be reliably described at this resolution. However, if there is a sufficient electron density in a given region and a corresponding hydrogen bond donor-acceptor pair in the model, this suggests the possible presence of such an interaction.

Our answer to reviewer #2 comments:

We are fully aware that the use of a C-terminally truncated construct limits this study due to the presumed role of the C-terminus in ASK1 dimerization. A C-terminally truncated construct consisting of TBD, CRR, and KD (residues 88-973) was used due to the low expression yield and solubility of full-length human ASK1.